# Reentrance of interface superconductivity in a high-$T_c$ cuprate heterostructure

J. Y. Shen[1,2,3,4,8], C. Y. Shi [1,8], Z. M. Pan[3], L. L. Ju[1], M. D. Dong[1,2,3,4], G. F. Chen[1,2,3,4], Y. C. Zhang[1,2,3,4], J. K. Yuan[3], C. J. Wu [3,4,5,6,7], Y. W. Xie [1] & J. Wu [2,3,4] ✉

Increasing the carrier density in a Mott insulator by chemical doping gives rise to a generic superconducting dome in high temperature superconductors. An intriguing question is whether a second superconducting dome may exist at higher dopings. Here we heavily overdope $La_{2-x}Sr_xCuO_4$ ($0.45 \leq x \leq 1.0$) and discover an unprecedented reentrance of interface superconductivity in $La_{2-x}Sr_xCuO_4$ /$La_2CuO_4$ heterostructures. As $x$ increases, the superconductivity is weakened and completely fades away at $x = 0.8$; but it revives at higher doping and fully recovers at $x = 1.0$. This is shown to be correlated with the suppression of the interfacial charge transfer around $x = 0.8$ and the weak-to-strong localization crossover in the $La_{2-x}Sr_xCuO_4$ layer. We further construct a theoretical model to account for the sophisticated relation between charge localization and interfacial charge transfer. Our work advances both the search for and control of new superconducting heterostructures.

The occurrence of a superconducting dome as a function of carrier density is a hallmark of high-temperature superconductivity[1,2]. As the carrier density is increased by chemical doping, cuprates evolve from antiferromagnetic insulators to superconducting strange metals, and eventually to non-superconducting metal. The critical doping levels corresponding to the two ends of the superconducting dome show little dependence on chemical constituents or lattice structures and thus are believed to be generic to cuprate superconductors. Such a dome-shape dependence of superconductivity on carrier density is universal to other families of superconductors as well, such as iron-based superconductors[3], heavy-fermion superconductors[4], twisted layer graphene[5], nickelate superconductors[6,7], etc. However, whether superconductivity can take place at higher dopings outside of the superconducting dome is an intriguing possibility worth exploring. Examples have been reported in the heavily overdoped $Sr_2CuO_{4-y}$[8,9] and $Ba_2CuO_{4-y}$[10]. Nevertheless, the reentrance of superconductivity has not been observed yet by tuning the carrier density in cuprate superconductors.

The discovery of interface superconductivity[11–24] provides a novel opportunity to address the relationship between superconductivity and carrier density from a unique perspective. More and more oxide heterostructures are found to host interface superconductivity, e.g., $LaAlO_3$/$SrTiO_3$[11,12], $La_{1.55}Sr_{0.45}CuO_4$/$La_2CuO_4$[13–15], $CaCuO_2$/$SrTiO_3$[16], $Ba_{0.8}Sr_{0.2}TiO_3$/$La_2CuO_4$[17], $La_2CuO_4$/$PrBa_2Cu_3O_7$[18], $EuO$/$KTaO_3$[19], and $LaAlO_3$/$KTaO_3$[19–21]. While the superconductivity is shown to reside at an interface layer[14], the number of charges at the interface determines the superconducting temperature $T_c$, which can be tuned by controlling the charge density in the parent materials through the charge transfer mechanism. Driven by the chemical potential difference, the charges redistribute across the interface of two-parent materials to achieve the chemical potential balance, giving rise to a superconducting interfacial layer. This process was modeled by using Poisson equations[15,25,26] and verified by the resonant soft X-ray scattering measurements[27]. Thus, modulating the doping levels of the parent materials effectively modulates the interfacial charge density and concomitantly the interface superconductivity.

[1]School of Physics, Zhejiang University, Hangzhou 310027, China. [2]Research Center for Industries of the Future, Westlake University, Hangzhou 310024, China. [3]Department of Physics, School of Science, Westlake University, Hangzhou 310024, China. [4]Key Laboratory for Quantum Materials of Zhejiang Province, School of Science, Westlake University, Hangzhou 310024, China. [5]New Cornerstone Science Laboratory, Department of Physics, School of Science, Westlake University, 310024 Hangzhou, China. [6]Institute for Theoretical Sciences, Westlake University, Hangzhou 310024 Zhejiang, China. [7]Institute of Natural Sciences, Westlake Institute for Advanced Study, Hangzhou 310024 Zhejiang, China. [8]These authors contributed equally: J. Y. Shen, C. Y. Shi. ✉e-mail: wujie@westlake.edu.cn

We study the heterostructure consisting of La$_2$CuO$_4$ (LCO) and the heavily overdoped La$_{2-x}$Sr$_x$CuO$_4$ (LSCO) ($0.45 \leq x \leq 1.0$) and discover a shocking reentrance of interface superconductivity as the chemical doping $x$ in LSCO increases. This cannot be explained by the electrostatic charge transfer model, cation interdiffusion or other innovative mechanisms, such as epitaxial-induced lattice distortion[28] and proximity-enhanced phase stiffness[29]. Instead, we elucidate that the anomalous reentrance of interface superconductivity is a manifestation of the sophisticated interplay between the charge localization in overdoped LSCO and the interfacial charge transfer. Though dynamic factors, such as charge mobility, have been overlooked so far in the charge transfer process due to lack of experimental evidence, here they play a dominant role. LSCO ($0.45 \leq x \leq 1.0$) is distant from the superconducting dome and its property is less explored due to the difficulty in synthesis and the lack of high-quality samples. It is widely accepted that the Fermi liquid behavior is recovered at the overdoped side of LSCO as superconductivity diminishes[2]. Increasing the doping further from $x = 0.45$ toward 1.0, however, makes LSCO appear insulating[30–32]. The underlying mechanism for this transition needs more thorough studies for the Zhang-Rice singlet[33], which is formed by a copper $3d_{x^2-y^2}$ hole hybridized with a hole on its neighboring oxygen $2p_{x,y}$ orbitals, does not apply for $x > 0.5$. Therefore, LSCO/LCO ($0.45 \leq x \leq 1.0$) heterostructure is of particular interest for the studies of high-temperature superconductivity and gives us an opportunity to construct a unified picture of the interface superconductivity in both metal/insulator and insulator/insulator heterostructures. This provides clues to guide us on the search and control of new interface superconducting heterostructures.

The epitaxial growth of LSCO ($0.45 \leq x \leq 1.0$) is technically challenging since the oxygen vacancy is prone to take place in the CuO$_2$ plane as the doping increases. In order to maximize the oxidating power and reduce the oxygen deficiency, the LSCO layer was grown within an oxidative environment with a mixture of ozone (10%) and oxygen (90%) gas kept at $3 \times 10^{-2}$ mbar. Both LSCO (26 nm) single layers and LSCO (26 nm)/LCO (13 nm) bilayers were epitaxially grown on the LaSrAlO$_4$ (001) substrate by pulsed laser deposition. The growth recipe has been optimized based on many rounds of growth-characterization cycles to produce correct oxygen stoichiometry and high-quality crystallinity as confirmed by the in situ Reflection High Energy Electron Diffraction (RHEED) and out situ X-ray diffraction (XRD) spectrum. The samples were patterned into a Hall bar shape by UV-photolithography for accurate transport measurement (more details on film synthesis, structural characterization, and transport measurement can be found in the Methods section and the Supplementary Information).

## Results and discussion

The sheet resistance $R_\square(T)$ for LSCO/LCO bilayers is shown in Fig. 1a as a function of $x$. Since the interfacial layers of LCO become conducting with holes transferred across the interface from the LSCO layer, the sheet resistance is a more suitable quantity to characterize electric transport than resistivity. The interface superconductivity manifests itself clearly in $R_\square(T)$ for $x = 0.45$ but gradually weakens as $x$ increases and diminishes completely for $x = 0.8$ at which no drop of resistance is present in its $R_\square(T)$. Then superconductivity revives as $x$ increases further to $x = 0.9$ and fully recovers till $x = 1.0$. It would be interesting to study how the interface superconductivity behaves beyond $x = 1.0$ but regrettably, the quality of the LSCO film degrades so much for $x > 1.0$ that it becomes polycrystalline.

The superconducting critical temperature $T_c$, extracted from $R_\square(T)$, unambiguously shows drastic non-monotonic dependence on $x$ (Fig. 1b). Three different criteria of $T_c$ yield qualitatively the same trend, evidencing that such an $x$-dependence does not rely on the choice of $T_c$ criterion. Therefore, the suppression of $T_c$ around $x = 0.8$ and the reentrance of $T_c$ at $x = 0.9$ is intrinsic to the LSCO/LCO interface superconductivity.

The XRD spectra taken from the LSCO/LCO bilayer show no meaningful difference between $x = 0.8$ and other doping levels in terms of the peak intensity and width of the bilayer XRD peak (see the Supplementary Information for details), ruling out the crystalline quality as the cause of the loss of interface superconductivity. Another possible cause is oxygen vacancies that are likely to be present for heavily overdoped LSCO films[30]. Although it is difficult to make a direct measurement of the density of oxygen vacancies in our films, it is known from previous studies that the interface superconductivity actually resides at an interfacial layer at the undoped LCO side[14] so the oxygen vacancies in LSCO layers should not have a direct effect on superconductivity. The same argument applies to other factors as well, such as the La/Sr disorder. Moreover, should factors like the oxygen vacancies, La/Sr disorder, and interfacial cation interdiffusion, impair the interface superconductivity in some ambiguous ways, it would affect the LSCO/LCO ($x = 1.0$) bilayer more severely than the LSCO/LCO ($x = 0.8$) bilayer−yet the former is superconducting, but the latter is not. Thus, we can exclude these factors as the major cause behind the scenes.

To find clues to the unusual $T_c(x)$ dependence, we plot $R_\square(T)$ of the LSCO/LCO bilayer with $R_\square(T)$ of the LSCO single layer at the same doping level for a one-to-one comparison (Fig. 2). The thickness of the LSCO layer in LSCO/LCO bilayers and LSCO single layers are exactly the same. Suppose there is no interfacial charge transfer, then the addition of the top insulating LCO layer should not change the electric

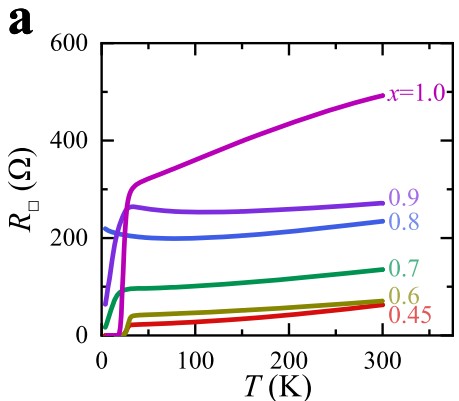

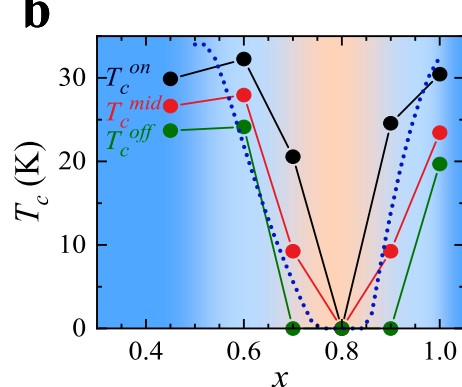

**Fig. 1 | Interface superconductivity in LSCO($x$)/LCO bilayers. a** The drop in the sheet resistance $R_\square(T)$ at low temperatures manifests the interface superconductivity. **b** Three criteria of the superconducting temperature $T_c^{on}$, $T_c^{mid}$, and $T_c^{off}$, corresponding to the 90%, 50%, and 10% of the sheet resistance in the normal state respectively, show anomalous non-monotonic dependence on the chemical doping $x$. The dashed line is calculated from the theoretical model proposed in this paper.

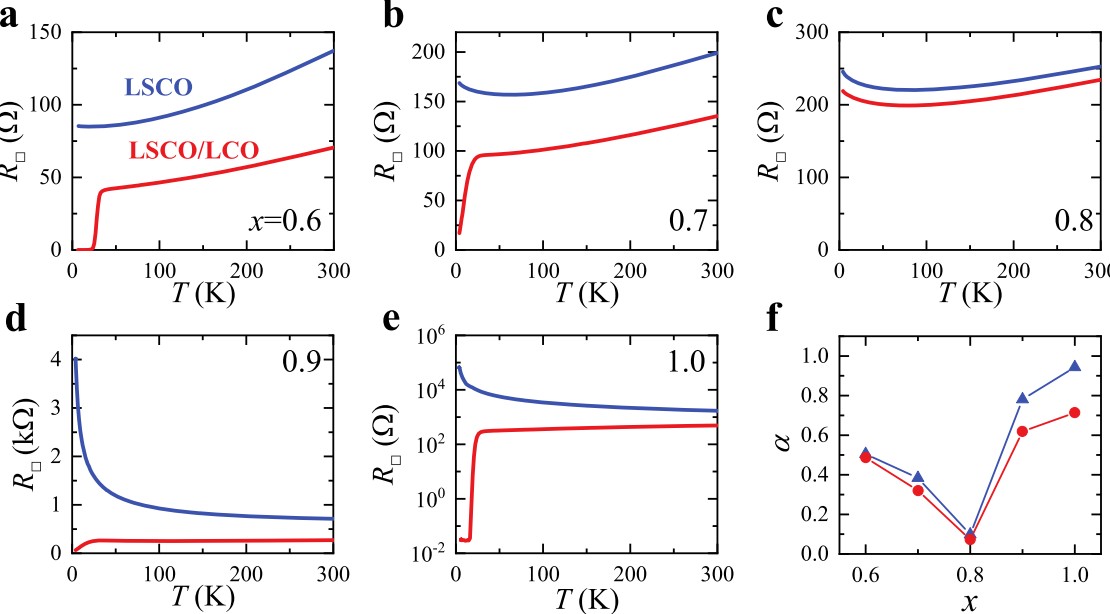

**Fig. 2 | Comparison of the sheet resistance $R_\square(T)$ from the LSCO($x$) single layer and the LSCO($x$)/LCO bilayer as an indicator for the interfacial charge transfer.** The thickness of the LSCO layer is kept at 26 nm, identical for all the films and the bilayers. The addition of the insulating LCO layer on the top doesn't contribute to the electric transport assuming no interfacial charge transfer. Thus the change in $R_\square(T)$ with/without the top LCO layer is a direct reflection of the charge transfer. **a–e** The difference in $R_\square(T)$ between the single layer and the bilayer is strongly $x$-dependent. **f** The ratio $\alpha \equiv (R(\text{LSCO}) - R(\text{LSCO/LCO}))/R(\text{LSCO})$ at two representative temperatures $T = 50$ K (blue triangles) and 300 K (red dots) is an indicator of the interfacial charge transfer and $\alpha(x)$ mimics the trend of $T_c(x)$ in Fig. 1b, which is no coincidence.

transport such that $R_\square(T)$ of the LSCO/LCO bilayer would be equal to that of the LSCO single layer. With the interfacial charge transfer, some LCO layers are doped with mobile holes and become conducting. The concomitant resistivity drop in LCO dominates over the increase of resistivity in LSCO layers due to the loss of carriers being transferred away, resulting in the decrease of the overall resistivity. Hence the change in $R_\square(T)$ is an indicator of the interfacial charge transfer. In stark contrast to one's naive expectation, the difference between $R_\square(T)$ of the LSCO/LCO bilayer and that of the LSCO single layer shows an anomalous doping dependence: The difference is smallest for $x = 0.8$ while it is significantly much larger for all other dopings (Figs. 2a–e). To be more quantitative, we calculate the ratio $\alpha = \Delta R_\square / R_\square(\text{LSCO})$, where $\Delta R_\square = R_\square(\text{LSCO}) - R_\square(\text{LSCO/LCO})$, and plot it as a function of $x$ for two representative temperatures (50 K and 300 K) (Fig. 2f). $\alpha(x)$ for both temperatures show very similar $x$-dependence, implying the charge transfer is weakly temperature-dependent. $\alpha$ decreases as $x$ increases till it reaches a minimum close to zero at $x = 0.8$. Then the trend is reversed and $\alpha$ increases with $x$ for $x > 0.8$. $\alpha(x)$ mimics $T_c(x)$ in Fig. 1b, indicating a generic connection between the charge transfer and the interface superconductivity.

This connection is also supported by the Hall effect measurements on both the LSCO/LCO bilayers and the LSCO single layers (Fig. 3). Although the Hall effect on the LSCO/LCO bilayers is complicated by the fact that with charge transfer the interfacial LCO layers are conducting and contribute differently from the LSCO layer to the overall Hall signal, we here simply use the change in the sheet Hall coefficient $R_H(T)$ with/without the top LCO layer as an indicator of the interfacial charge transfer—an idea similar to the comparisons of resistivity in Fig. 2. It is truly remarkable that such a change in $R_H(T)$ is substantial for all dopings except $x = 0.8$ (Fig. 3), again showing the charge transfer is severely suppressed at this doping. In addition, it should be noted that $R_H$ for the LSCO single layers is strongly doping- and temperature-dependent such that $R_H$ changes sign from positive to negative, and then back to positive with increasing $x$. This probably is related to the non-trivial evolution of LSCO Fermi surface that is out

of the scope of the current work and deserves elaborate investigations and discussions in a separate paper.

Apparently, the LCO layer would not be superconducting without the charges transferred from the LSCO layer. This explains the demise of interface superconductivity at $x = 0.8$. However, the more profound question is why the charge transfer between LSCO and LCO has such an anomalous dependence on the doping level $x$ of the LSCO layer.

The charge transfer is driven by the difference in the chemical potentials of LCO and LSCO, so it is worth checking whether the chemical potential of LSCO around $x = 0.8$ has any anomaly. For this purpose, we used the Kelvin probe force microscopy (KPFM) to directly measure the LSCO work function (see the Methods section for more details). The previous studies showed that the work function of LSCO remains constant for $0 < x < 0.16$ and then it increases linearly with $x$ for $0.16 < x < 0.45$[15,34]. The KPFM measurements show the linear $x$-dependence of the work function continues for $0.45 \leq x \leq 1$ (Fig. 3a). Thus, it is truly abnormal that the charge transfer around $x = 0.8$ is suppressed despite the big difference in the work function between LSCO and LCO.

It should be pointed out that there is a fundamental difference between the LSCO/LCO interface and the LaAlO$_3$/SrTiO$_3$ (or KTaO$_3$) interface[11,12,19–21]. LaAlO$_3$ and SrTiO$_3$ are both insulators so the two-dimensional electron gas at the interface, once formed due to interfacial charge transfer, is confined to the interface due to the existence of the bandgaps in LaAlO$_3$ and SrTiO$_3$. The effective potential at the interface forms a quantum well and gives rise to energy levels and sub-bands. The charge transfer in LaAlO$_3$/SrTiO$_3$ is determined by the filling of these sub-bands[35–37]. In stark contrast, overdoped LSCO is metallic (though the effect of localization gets stronger as $x$ approaches 1.0, see Figs. 2a–e and the Supplementary Information for details). The missing of a bandgap at the LSCO side means electrons are not confined at the LSCO/LCO interface. Moreover, δ-doping experiment[14] clearly showed that superconductivity resides at the interfacial CuO$_2$ plane located at the LCO side so the main role of the overdoped LSCO is to provide the transferred charges. Therefore, it is essential to find

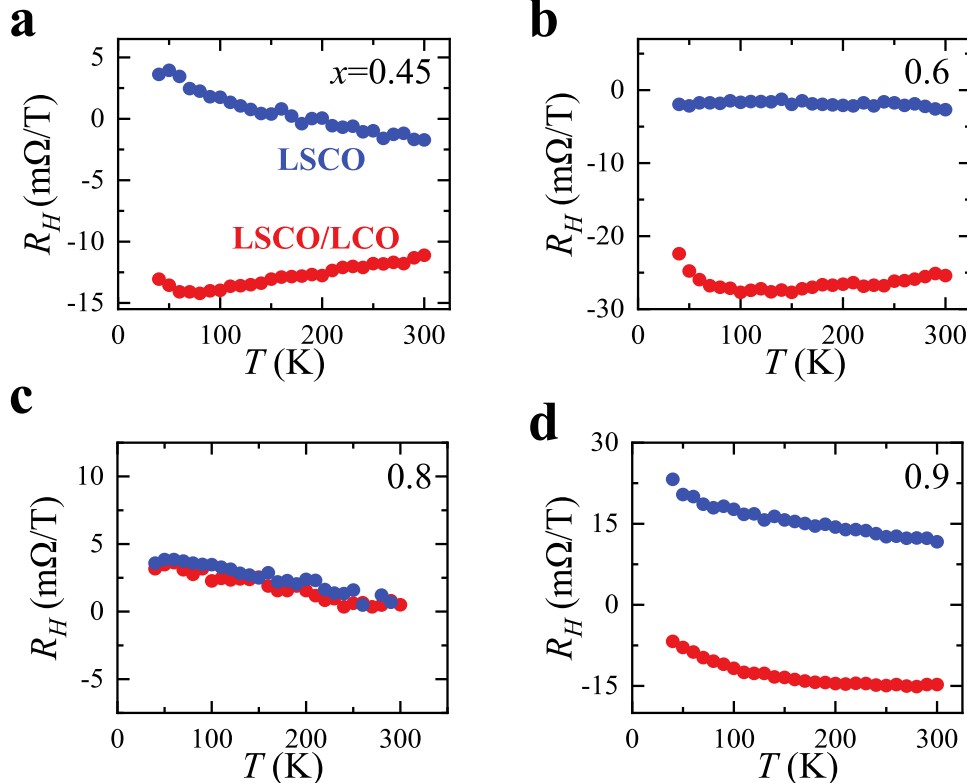

**Fig. 3 | Comparison of the sheet Hall coefficient $R_H(T)$ from the LSCO($x$) single layer and the LSCO($x$)/LCO bilayer. a–d** The change in $R_H(T)$ with/without the top LCO layer is another reflection of the interfacial charge transfer. At $x = 0.8$, the change is the smallest, in agreement with $\alpha(x)$ in Fig. 2f.

the underlying mechanism that suppresses interfacial charge transfer around $x = 0.8$.

The key is to go beyond the normal electrostatic approach and consider the dynamic processes in charge transfer. The chemical potential difference caused by the application of an external voltage does not necessarily give rise to a sizable current if the mobility of charges in the material is very low, e.g., the charge flow is blocked by the Anderson localization[38–40], or, Mott localization mechanisms[41,42]. Following this line of thinking, we focused on the low-temperature electric transport properties of the LSCO single layers (Figs. 4b–d). For LSCO ($x = 0.45$), its longitudinal resistivity $\rho(T)$ increases monotonically with $T$, manifesting the typical metallic behavior. In contrast, $\rho(T)$ of LSCO ($x = 0.7$) and LSCO ($x = 0.8$) shows a minimum at $T \sim 60$ K and 80 K, respectively, and then it increases as $T$ lowers further. The low-temperature $\rho(T)$ conforms to the logarithmic relation $\rho(T) \propto \ln(\frac{1}{T})$ (Fig. 4b). To examine whether this is related to the Kondo effect[43], we measured the magneto-resistivity $\frac{\triangle\rho(B)}{\rho(0)} \equiv \frac{\rho(B)-\rho(B=0)}{\rho(B=0)}$ and found it significantly deviates away from the relation $\frac{\triangle\rho(B)}{\rho(0)} \propto -B^2$, inconsistent with the prediction based on the Kondo effect[44]. Instead, the logarithmic $\rho(T)$ and the magneto-conductivity can be well fitted by the weak localization theory[40,45,46] (Fig. 4c), in which the localization is caused by the interference of electron wavefunctions during propagations in the presence of impurity and defects. Note that a similar $\rho(T) \propto \ln(\frac{1}{T})$ dependence was also observed in underdoped LSCO ($x = 0.048$)[47–49] but the corresponding magneto-transport showed different behaviors from those in Fig. 4c so the underlying mechanisms are different for under- and overdoped LSCO. As $x$ reaches 1.0, the logarithmic $T$-dependence breaks down and $\rho(T)$ appears insulating from room temperature down to low temperatures. However, $\rho(T)$ for LSCO ($x = 1.0$) is best fitted by the expression $\rho(T) = \rho_0 \exp\left(\left(\frac{T_0}{T}\right)^{1/4}\right)$

(Fig. 4d), derived from the three-dimensional variable-range hopping (VRH)[42]. Here $\rho_0$ and $T_0$ are two fitting parameters. Concomitantly, the VRH expression also fits nicely the Hall coefficient of LSCO ($x = 1.0$) $R_H(T) = A_H \exp\left(\left(\frac{T_H}{T}\right)^{1/4}\right)$ with $A_H$ and $T_H$ being the fitting parameters (the inset of Fig. 4d)[50].

To conclude the electric transport properties of the LSCO single layers, we see a general trend that the localization becomes stronger as $x$ increases. Using the kink in $\rho(T)$ as an indicator, the weak localization behavior emerges at $T < T^*$ (~25 K) for $x = 0.6$ and the corresponding $T^*$ increases to 60 K ($x = 0.7$) and 80 K ($x = 0.8$). Meanwhile, the density of the localized states increases at the cost of that of itinerant states, which is reflected by the rapid increase in resistivity with $x$ at $T > T^*$. For $x > 0.8$, all charges become localized, and the weak localization behavior eventually evolves into the VRH behavior. The plausible sources for the charge localization are the La/Sr disorder and oxygen vacancies, which both occur more often as the doping $x$ increases.

The interface superconductivity is weakened as the charge localization in the LSCO layer becomes more prominent. This is no coincidence. The charge localization suppresses the interfacial charge transfer and eventually eliminates the superconductivity in the LCO layer as the suppression becomes strong enough at $x = 0.8$. As the doping increases further to $x = 1.0$, the localization becomes even stronger; however, the chemical potential difference between LSCO and LCO also increases and the total doped charge in the bilayer grows in population. When the latter factor dominates, charges are forced to move across the interface in a manner similar to the electric breakdown in insulators, then the interface superconductivity revives at $x = 0.9$ and 1.0.

Based on the above insights, we construct a model to numerically calculate the charge transfer under the influence of charge

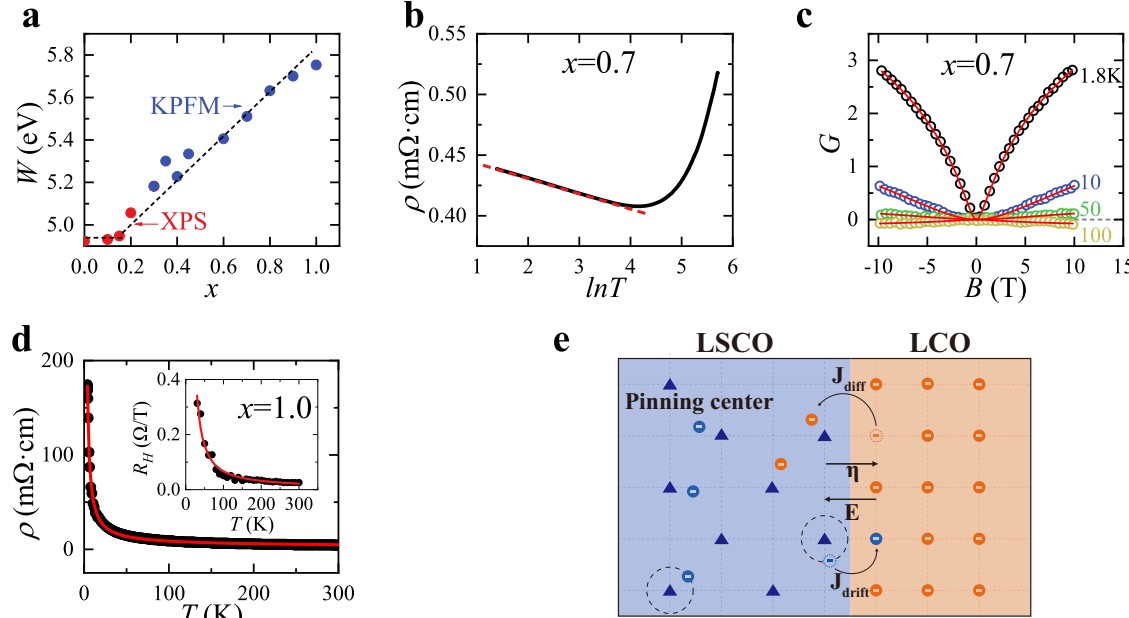

**Fig. 4 | Charge localization and its determinant role in the interfacial charge transfer. a** The work function $W(x)$ from the XPS[15,34] and our KPFM measurements. **b** For the LSCO($x = 0.7$) single layer, $R_\square(T)$ shows $lnT$ dependence at low temperatures (red dashed line), suggestive of the weak localization behavior. **c**, Its magneto-conductivity (open circles) $G = \frac{2\pi^2\hbar}{e^2}(\sigma(B) - \sigma(B=0))$ can be well fitted (red solid lines) by an expression $G = A \times \frac{3}{2}\left[\psi\left(\frac{1}{2} + \frac{B_1}{B}\right) - \ln\left(\frac{B_1}{B}\right)\right]$ derived from the weak localization theory[46]. Here $A$ and $B_1$ are temperature-dependent fitting parameters and $\psi$ is the digamma function (PolyGamma function of order 0). **d** For the LSCO($x = 1.0$) film, $R_\square(T)$ (black dots) and the Hall coefficient $R_H(T)$ (the inset) can both be well fitted by the VRH expressions (red line) under the strong localization limit[42,50]. **e** A schematic cartoon illustrates two most important processes in determining the interfacial charge transfer in the presence of strong localization: Charge drift and charge diffusion. $\eta$ is the net transferred charge and $E$ is the corresponding built-in electric field. The hopping of charges from LSCO to LCO is suppressed by the localization field from the pinning centers in the LSCO layer.

localization. It includes two counter-processes that reach dynamic balance at equilibrium (Fig. 4e). One is the diffusion current from the LSCO layer to the LCO layer $J_{diff} = eD\frac{dp}{dz} = \mu k_B T \frac{dp}{dz}$. Here $e$ is the electron charge, $D$ is the diffusion coefficient, $\mu$ is the charge mobility, $k_B$ is the Boltzmann constant, $p$ is the mobile charge density and the $z$-direction is normal to the film. The other is the drift current from the LCO layer to LSCO layer, which is $J_{drift} = e\mu p E$ for $x \leq 0.8$. The built-in electric field $E$ is generated by the charges $\eta$ transferred across the interface so $E = \frac{e}{\varepsilon a^2}\eta$, where $\varepsilon$ is the dielectric constant and $a$ is the in-plane lattice constant. The condition for equilibrium requires $J_{drift} = J_{diff}$. Retrieving the parameters from the transport of the single-layer LSCO, we solved for $\eta(x)$. The quantitative results confirm that the suppression of charge transfer for $0.45 \leq x \leq 0.8$ is mainly due not to the weak localization but to the reduction of mobile charges owing to the disorder-induced pinning effect. For the more overdoped bilayer LSCO/LCO ($0.8 < x \leq 1.0$), all the charges are localized and both $J_{diff}$ and $J_{drift}$ are embodied by VRH of charges[51]. Thus, the equilibrium is achieved by the balance of chemical potentials $u_L - \Delta u = u_R + \Delta\phi$, where $u_L$, $u_R$ are the chemical potentials of LSCO and LCO respectively. The charge transfer causes $u_L$ to decrease by the amount $\Delta u$ with the relation $\eta = \int_{u_L - \Delta u}^{u_L} N(E')dE'$. Here $E'$ is the energy and $N(E')$ is the density of the state of LSCO. Meanwhile, the change in the potential is $\Delta\phi \approx eE\frac{c}{2}$, where $c$ is the out-of-plane lattice constant. Then we approximated the distribution of the local pinning potentials in LSCO to be a Gaussian distribution so $N(E') = \frac{N_0}{\sqrt{2\pi}\sigma}\exp\left(-\frac{(E' - u_0)^2}{2\sigma^2}\right)$ with $N_0$, $\sigma$, $u_0$ being the fitting parameters. By combining these relations together, we solved for $\eta(x)$ for $0.8 < x \leq 1.0$. The calculation confirms that there are two competitive energy scales affecting interfacial charge transfer. One is the chemical potential difference between LSCO and LCO, and the other is the strength of the local pinning potentials. When the former

factor dwarfs the latter one as $x$ increases, the charge transfer revives even in the presence of charge localization.

Then $T_c$ of the interface superconductivity can be obtained by the empirical relation $T_c = 3200 \times (p - 0.06) \times (0.26 - p)$ where $p$ is the carrier density at the interfacial $CuO_2$ plane with the highest $T_c$ (here the maximum $T_c$ is 32 K, lower than that of LSCO/LCO bilayer synthesized by oxide-MBE due to rougher interface). The calculated $T_c(x)$ dependence reproduced the trend from the experimental data very nicely (the dashed line in Fig. 1b), showing that the interface superconductivity indeed can be completely suppressed at $x = 0.8$ but fully recovers at $x = 1.0$ as charge localization and carrier density increases (see the Supplementary Information for more details on the theoretical modeling).

It is noted that $R_\square(T)$ curves for LSCO/LCO ($x = 0.7$ and 0.9) show border transition. Though the superconducting transition of copper oxide superconductors is generically broader than conventional BCS superconductors due to vortex excitations, it is very likely that the broadening is also related to inhomogeneity in the LSCO/LCO bilayers, which originates from La/Sr randomness, oxygen vacancies, and spontaneous electron phase separation[52,53]. In the presence of inhomogeneity, there is a spread of charge density and mobility in different regions of the bilayer. The superconducting transition occurs through percolations of superconducting clusters. This effect is insignificant when $T_c$ is independent or weakly dependent on the charge density, like in LSCO/LCO ($x \leq 0.6$). The broadening of transition is most noticeable for LSCO/LCO with marginal dopings ($x = 0.7$ and 0.9), at which the effective dopings of some clusters are closer to $x = 0.8$ with much lower or diminished $T_c$.

One monolayer $CuO_2$ deposited on a $Bi_2Sr_2CaCu_2O_{8+\delta}$ substrate (BSCCO/$CuO_2$) was found to superconduct with nodeless pairing[54]. Due to charge transfer from BSCCO, the top $CuO_2$ layer is

heavily overdoped according to DFT calculation. An extended $s$-wave pairing symmetry[55,56] is suggested to be a result of the emergence of $d_{3z^2-r^2}$ orbit for doping levels $x > 0.8$. While this interface superconductivity bears some apparent similarity to our results, we notice a direct comparison between them shows significant differences as well. Our measurements on LSCO ($0.45 \le x \le 1.0$) single layers show no sign of the existence of a second superconducting dome at heavily overdoped region. And $T_c$ of LSCO/LCO, which is determined by the highest $T_c$ from one of the interfacial $CuO_2$ planes at the LCO side, never exceeds $T_c$ of the optimally doped LSCO single layer ($x = 0.16$), indicating the consistency in their superconducting mechanisms. Nevertheless, whether the $d_{3z^2-r^2}$ band enters the Fermi surface of heavily overdoped LSCO and the possible role it may play in LSCO/LCO interface superconductivity are intriguing questions that invoke more thorough experimental studies and DFT calculations. This will clarify if other conditions, e.g., epitaxial tension, surface reconstruction, or removal of the top La-O plane, might be relevant in producing the peculiar superconducting state in BSCCO/$CuO_2$.

Our results illustrate that the interfacial charge transfer is not solely determined by the difference in chemical potentials between two parent compounds. Instead, the electric transport properties of compounds, such as the mobility of carriers, also play a vital role. This work shines a light on the long-standing puzzle of whether a (super)conducting interfacial layer can form in certain heterostructures and provides a guideline for searching for novel materials with interface superconductivity. By including the charge drift and diffusion under the influence of charge localization, our numerical model is a step forward toward a practical theory with prediction capabilities of the interface superconducting $T_c$.

## Methods

### Film synthesis, lithography, and characterization

Heavily overdoped LSCO($0.45 \le x \le 1.0$) single layers and LSCO/LCO($0.45 \le x \le 1.0$) bilayers were synthesized on LaSrAlO$_4$(001) substrates by the pulsed laser deposition technique. The substrate was pre-annealed in situ at 800 °C under $3 \times 10^{-4}$ mbar oxygen for 20 min to achieve a clean surface and good morphology. A KrF excimer laser (248 nm) with the laser fluence 1.2 J/cm$^2$ was used. The laser pulse frequency was set to 4 Hz. During growth, the substrate temperature was kept at 730 °C at $3 \times 10^{-2}$ mbar within a mixture of ozone (10%) and oxygen (90%). An in situ reflection high-energy electron diffraction (RHEED) instrument was used to monitor the real-time growth and the clear oscillations in the intensity of RHEED spots confirm the high crystalline quality and nice layer-by-layer growth of LSCO and LCO. The thickness of LSCO is 26 nm (equivalent to 20 unit cells) for both the LSCO single layer and LSCO/LCO bilayer. An addition of a 13 nm LCO layer was deposited on top of the LSCO/LCO bilayer. After growth, the films were cooled to 400 °C with a rate of 100 °C/min and held for 15 min at pressure gradually rising from $3 \times 10^{-2}$ to 4.5 mbar in mixed ozone/oxygen, followed by a second annealing at ~200 °C for 30 min within $2 \times 10^2$ mbar oxygen. This growth recipe is the result of optimizations based on many rounds of growth-characterization cycles, so it minimizes the oxygen vacancies in the overdoped LSCO layer and removes excess interstitial oxygen from the undoped LCO layer.

The films were patterned by standard UV-lithography and ion milling to form Hall bar devices. 50 nm gold was deposited onto the contact pads for good Ohmic contact. The distance between two contact pads is 900 μm for measurements of sheet resistance.

The $c$-axis lattice constants of the films were measured by the XRD method on a Bruker AXS D8-Discover with Cu Kα radiation ($\lambda = 1.5406$ Å). A detailed analysis of the XRD measurements is included in the Supplementary Information.

### Transport measurements

For resistance and Hall effect measurements, the samples were mounted onto a cryocooler to reach temperatures as low as 4 K and a magnetic field as high as 1 T. For magneto-resistance measurements, the samples were loaded into a Physical Property Measurement System (Quantum Design, PPMS dynacool) to reach 1.8 K and 9 T magnetic field. Keithley 6221 sourcemeters and Keithley 2182A nanovoltmeters were used to generate DC current and measure the corresponding DC voltages.

### Work function measurements

Kelvin probe force microscopy (Oxford instrument Cypher ES) was employed to measure the work function of the samples. A 50 nm gold film grown on a LaSrAlO$_4$(001) substrate was measured with the samples for calibration of the instrument. The values of work function were taken at multiple well-separated locations on the same sample and then averaged to reduce the influence of defects and inhomogeneity.

## Data availability

The data that support the findings of this study and all other relevant data are available from the corresponding author upon request.

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

## Acknowledgements

This work was supported by the Research Center for Industries of the Future (RCIF project No. WU2023C001 to J.W.) at Westlake University, the National Natural Science Foundation of China (Grant No. 12174318 to J.W., 12325402 to Y.W.X., 12074334 to Y.W.X., 12234016 to C.J.W., and 12174317 to C.J.W.), the Zhejiang Provincial Natural Science Foundation of China (Grant No. XHD23A2002 to J.W.), the New Cornerstone Science Foundation to C.J.W. and the Westlake Multidisciplinary Research Initiative Center Award (Grant No. MRIC20210102 to J.W.). We acknowledge the assistance provided by the Westlake Center for Micro/Nano Fabrication and the Instrumentation and Service Center for Physical Sciences at Westlake University.

## Author contributions

The films were synthesized and characterized by Y.W.X., C.Y.S., and L.L.J. The lithography was done by J.Y.S. and M.D.D. The transport measurements were done by J.Y.S., G.F.C., and Y.C.Z. The theoretical modeling was done by Z.M.P., J.W., J.K.Y., and C.J.W. and the analysis and interpretation put forward by J.W.

## Competing interests

The authors declare no competing interests.
