## [Peer Review File · Nature Communications]

Reentrance of interface superconductivity in a high- T_c cuprate heterostructureREVIEWER COMMENTS

Reviewer #1 (Remarks to the Author):

The interfacial superconductivity in oxide heterostructures have been of interest with potential for plethora of quantum phenomena. The authors report re-entrant superconductivity at the LCO/LSCO interface. This behavior is concurrent with the suppression of the interfacial charge transfer and the weak-to-strong localization crossover in the LSCO. The authors, however, need to consider further characterization of the interface, considering other effects e.g., epitaxial strain. The xrd corresponding to re-entrant heterostructures, for example, does not exhibit fringes which could be sign for interfaces which are not abrupt. Authors should add the band structure explaining the charge transfer claimed. Interfacial superconductivity and charge transfer at recently discovered KTaO₃ interfaces cited by authors is an example showing charge transfer atomically scale characterized and charge transfer characterized by PS calculations (e.g., see LSMO/KTO Science Adv. 9 (7), eadf1414(2023), LMO/KTO Arxiv: 2304.14426 (2023), LCO/KTO APL 118 (19), 192905 (2021))

Reviewer #2 (Remarks to the Author):

The manuscript "Reentrance of interface superconductivity in a high-T_c cuprate heterostructure", by Jingyi Shen, Chuanyu Shi, Zhiming Pan, Lele Ju, Mingdong Dong, Gangfan Chen, Yichi Zhang, Congjun Wu, Yanwu Xie, and Jie Wu, contains result that are timely and interesting. However, before a definite statement about their relevance and soundness can be made, I ask the authors to consider the issues that are listed hereafter.

1. Usually, the quantum confinement in the direction perpendicular to the interface produces a series of levels (that are the thresholds of two-dimensional sub-bands associated with the motion parallel to the interface). A discussion about the physics of the interface (in particular, about its superconducting properties) should attempt a connection with this sub-band structure and the hierarchy of the sub-band filling, when the density of carriers at the interface is varied.
2. The solution of the Schroedinger-Poisson equations for oxide interfaces like LAO/STO or LTO/STO [PRL 116, 026804 (2016)] shows that the sub-band structure may be not rigid, and electron phase separation may occur, resulting in an inhomogeneous state at the interface. As I argue hereafter, I suspect that at least for some carrier density, the authors are observing the phenomenology of an inhomogeneous electron gas at the interface.
3. My main line of reasoning, when it comes to the homogeneity of the electron gas rests on the curves in Fig. 1a. The curves at 0.7 and 0.9 are so broad that their width cannot be possibly accounted for any reasonable fluctuational mechanism and rather points toward inhomogeneity [PRB 84, 014514 (2011)]. The fact that the resistance significantly drops, without seemingly reaching the zero-resistance state,

likely indicates that a sizable superconducting cluster exists in the system, but does not percolate. The percolative character of the superconducting transition again seems to point toward inhomogeneity.

4. The authors themselves write that their interfaces are structurally homogeneous, but admittedly cannot control oxygen vacancies. The argument that they should not have a direct effect on superconductivity is not compelling, my counterargument is that mobile oxygen vacancies may balance the inhomogeneous distribution of charge carriers at the interface, making electron phase separation easier. I point out that such an inhomogeneous electron gas need not be accompanied by a correspondingly inhomogeneous lattice configuration. The modulation of the electron charge may be mild (a tiny fraction of electron per unit cell), so that the lattice may be nearly unaffected.

5. I am also not convinced of the authors' discussion about the $\log(1/T)$ term in the resistivity. This phenomenology was found in underdoped cuprates long ago [PRL 77, 5417 (1996)], but it was convincingly argued that weak localization cannot possibly explain it [C. Castellani, P. Schwab, M. Grilli, (2002). On Localization Effects in Underdoped Cuprates. In: A. Bianconi, N. L. Saini (eds), Stripes and Related Phenomena. Selected Topics in Superconductivity, pp 361–367, vol 8. Springer, Boston, MA. https://doi.org/10.1007/0-306-47100-0_45]. The main argument is that weak localization gives a log contribution to the conductivity, to have a log contribution to the resistivity, it must be small so that a perturbative expansion can transform a log in the conductivity into a log in the resistivity. In underdoped cuprates, the log contribution is by no means small. Now, it would be somewhat surprising that there are two different mechanisms for the log contribution to the resistivity in bulk samples and in heterostructures.

A thorough discussion of the points listed above is mandatory.

There are also minor points:

- a. The authors write "caution interdiffusion", I suspect they mean "cation interdiffusion".
- b. The authors write "insert of Fig. 3d", they probably mean "inset of Fig. 3d".

Once the authors have responded to my criticism and complied with the issues I raised, I suggest that their manuscript may be reconsidered for publication.

Reviewer #3 (Remarks to the Author):

The paper "Reentrance of interface superconductivity in a high- T_c cuprate heterostructure" by J. Y. Shen addresses an extremely important and interesting issue, namely superconductivity in cuprates in the high doping regime ($x > 0.8$). Specifically, the authors investigate interfacial superconductivity in the LSCO/LCO system, where they have successfully measured a fascinating phenomenon, namely the reentrance of superconductivity in the highly overdoped regime $0.8 < x < 1.0$. To explain the observed

phenomenon, the authors propose a theoretical model based on charge transfer at the interface, which is a result of the interplay between the electrostatics of the junction determined by the work function in LSCO which changes with doping, and the charge localization effect in LSCO, experimentally confirmed by magnetconductance measurements. Undoubtedly, the research results contribute to a significant and novel direction in the study of high-temperature superconductivity.

Here are my comments and questions regarding the paper:

1. Undoubtedly, the paper presents a very interesting experimental result, namely the reentrance of superconductivity in the range of high doping. Although the authors state in the introduction, "the reentrance of superconductivity has not been observed yet by tuning the carrier density in cuprate superconductor.", I would like to point out that a similar phenomenon was observed in 2016 for a monoatomic CuO₂ layer deposited on a Bi₂Sr₂CaCu₂O_{8+δ} substrate (CuO₂/BSCCO) [Sci. Bull. 61, 1239 (2016)]. However, it does seem that these are indeed the two papers in which superconductivity has been observed in cuprates in the highly overdoped regime. Therefore, I consider the experimental result presented in this paper to be highly significant.
2. However, in my opinion, the theoretical model explaining the observed phenomenon is too simplified. In fact, it reduces to the standard procedure of determining the built-in potential in junctions, namely the balance between diffusion current and drift current. However, I do not fully understand how this model accounts for the localization and the change in its strength with increasing doping, which the authors mention when analyzing the resistance measurements. It seems much more reasonable in this regard to consider a full Schrödinger-Poisson model limited to a few relevant bands that participate in charge redistribution at the junction. Furthermore, I would like to point out that the entire paper lacks a model describing superconductivity and the critical temperature which in the paper is described by the empirical formula.
3. In the paper, the reentrance effect of superconductivity was explained based on charge transfer blockade for $x=0.8$. However, it seems that the presented explanation may not be the only valid one in this case. The considered system is somewhat similar to CuO₂/BSCCO [Sci. Bull. 61, 1239 (2016)], where increased doping leads to the appearance of a new band (Lifshitz transition), which is associated with the emergence of superconductivity. A comprehensive theoretical model of the superconducting state for this system was presented in Phys. Rev. B 103, 144511 (2021), where it was clearly shown that around $x=0.8$, a Lifshitz transition occurs (an additional $d_{(x^2-y^2)}$ band crosses the Fermi level), and the induced superconductivity has the extended s-wave symmetry. It should be noted that in the considered paper, a certain change in the electronic structure is observed for LSCO with increasing x - the sign change of the Hall resistance is observed, which the authors suggest may be evidence of a Fermi surface modification. Taking into account a global trend that occurs in the phase diagram of doped cuprates, it can be expected that the phenomenon of superconductivity emergence in the range of high doping described in Phys. Rev. B 103, 144511 (2021) also takes place in the LCO/LSCO system, especially considering that the doping ranges where superconductivity is observed in both papers are the same. Therefore, from a theoretical standpoint, it would be necessary to perform DFT calculations for LCO/LSCO to investigate the interface's influence on the energetic distribution of bands and create a model of superconductivity based on such calculations. Here, I would like to mention that the interface

can significantly alter the electronic structure of a material, as evidenced by the 50 meV lowering of the d_{xy} bands in the LAO/STO system. In conclusion, it appears that a thorough analysis of the electronic structure in LCO/LSCO would be necessary to create a complete model of the described phenomenon and confirm the feasibility of the proposed model in the paper.

4. It seems that a measurement of local density of states could serve as confirmation or refutation of the theoretical predictions from the PRB paper. In the case of extended s-wave superconductivity, we should observe a characteristic U-shaped profile, in contrast to the d-wave symmetry characterized by the V-shaped local density of states. Are such additional measurements possible in the proposed system?

5. A very basic question: Where is the superconductivity located? Is there a thin copper-oxygen layer near the interface on the LSCO side? Or is it rather located on the LCO side where the charge is transferred ?

In conclusion, while I find the experimental results interesting and significant for the field of high-temperature superconductors, I would recommend undertaking substantial efforts to develop a realistic theoretical model that adequately explains the observed phenomenon.

Point-by-point response to the referees' comments.

Reviewer #1 (Remarks to the Author):

Comment 1: The interfacial superconductivity in oxide heterostructures have been of interest with potential for plethora of quantum phenomena. The authors report re-entrant superconductivity at the LCO/LSCO interface. This behavior is concurrent with the suppression of the interfacial charge transfer and the weak-to-strong localization crossover in the LSCO.

Response 1: We thank the Referee very much for recognition of the value of our work.

Comment 2: The authors, however, need to consider further characterization of the interface, considering other effects e.g., epitaxial strain. The xrd corresponding to re-entrant heterostructures, for example, does not exhibit fringes which could be sign for interfaces which are not abrupt.

Response 2: To address the strain effect, we measured the in-plane and out-of-plane lattice constants of LCO/LSCO bilayers. Apparently, the in-plane lattices of the epitaxially grown LCO and LSCO layers are identical to each other and both roughly follow the in-plane lattice of the substrate LSAO(001) at three representative dopings (Table R1), which is expected under the conditions of epitaxial growth. The differences in a/b constant are within measurement uncertainty. Despite the drastic differences in T_c of the three dopings, the epitaxial tension remains approximately the same. Thus, it is safe to conclude that there are epitaxial strains exerted on LCO/LSCO bilayers but they are not the reasons for the reentrance of interface superconductivity. We've added this information to the Supplementary Information.

As for the sharpness of the interface, indeed no fringe peaks are present in the XRD spectrum at several dopings. We've repeated the growth a couple of times and found that T_c of LCO/LSCO bilayer is highly repeatable for every doping under study (with variations less than 3 K from sample to sample). We wish we could further improve the interface quality in the future but currently there is no experimental evidence or conceivable theory suggesting the reentrance of superconductivity is mainly caused by the morphology at interface since the interface quality of LCO/LSCO remains roughly the same for $0.45 \leq x \leq 1.0$ (Fig. S1b).

	a/b lattice constant (Å)	c lattice constant (Å)
LaSrAlO ₄ substrate	3.76	12.636
La _{1.4} Sr _{0.6} CuO ₄ /La ₂ CuO ₄	3.77	13.246
La _{1.2} Sr _{0.8} CuO ₄ /La ₂ CuO ₄	3.76	13.215
LaSrCuO ₄ /La ₂ CuO ₄	3.75	13.217

Table R1: The lattice constants of LSAO substrate and LSCO films, determined by XRD measurements.

Comment 3: Authors should add the band structure explaining the charge transfer claimed. Interfacial superconductivity and charge transfer at recently discovered KTaO₃ interfaces cited by authors is an example showing charge transfer atomically scale characterized and charge transfer characterized by PS calculations

Response 3: We thank the referee for bringing these nice references to our attention and we've added them to the revised manuscript. We realize that we haven't made it clear the difference of interface superconductivity between LCO/LSCO and LAO/STO or LAO/KTO.

1. LAO, STO, KTO are all insulators so the two-dimensional electron gas (2DEG) at the interface, once formed due to interfacial charge transfer, is confined to the interface by the bandgaps at LAO, STO or KTO. This quantum well type potential gives rise to energy levels and sub-bands. In stark contrast, overdoped LSCO is metallic (though the effect of localization gets stronger as x approaches 1.0, Fig. S2 in Supplementary Information). The missing of a bandgap at the LSCO side means **electrons are not confined at the LCO/LSCO interface.**

2. δ -doping experiment clearly showed that the interface superconductivity resides at the interfacial CuO_2 plane located at the LCO side (see Response 16 for detailed discussions) so **the main role of the overdoped LSCO is to provide the transferred charges.**

3. **The distribution of charges across the LCO/LSCO interface at equilibrium is determined by the detailed balance of two counter-processes:** the difference in work function between LCO and LSCO drives holes to flow from LSCO to LCO, and the built-in electric field eventually grows big enough to stop such a flow. Here the electric field is generated by the imbalance between positive and negative charges as LSCO loses holes and LCO gains holes.

This is very different from LAO/STO or LAO/KTO where the charge density is determined by the filling of sub-bands.

4. We take it as an empirical fact that doping LCO with holes gives rise to high temperature superconductivity and the quantitative relation between superconducting T_c and the hole density p is given by $T_c = 3200 \times (p - 0.06) \times (0.26 - p)$ (See Response 13 for more thorough discussions on this).

5. T_c of interface superconductivity is dictated by one of the CuO_2 planes at LCO side that is closest to the optimal doping $p = 0.16$ and therefore has highest T_c among all the CuO_2 planes.

6. The above scheme of modeling charge transfer and interface superconductivity is quite successful for LCO/LSCO ($x \leq 0.45$) with calculated $T_c(p)$ in good agreement with experiments (see Reference 15, 25-27 of the manuscript).

7. The existing scheme, however, cannot explain the reentrance of interface superconductivity unless we take into account the effect of charge localization on interfacial charge transfer. Thus, we developed a new model to calculate charge transfer and $T_c(p)$ under the influence of charge localization, and found it captures the essence of the reentrance, as we have shown in the manuscript.

As for point 3 on the modeling of charge transfer, we have tried the Schrodinger-Poisson equation approach as the Referee suggested. The results show no fundamental difference with our original Poisson equation approach (Fig.R1). Thus, without taking into account the influence of charge localization, the Schrodinger-Poisson model cannot explain the reentrance of superconductivity as well. We have added the above discussions to the manuscript and the Supplementary Information. We hope this is crystal clear to our readers now.

Fig. R1: The calculated charge distribution from Schrodinger-Poisson equation (a) and Poisson equation (b) for LCO/LSCO ($0.3 \leq x \leq 0.9$) without taking into account the influence of charge localization. The value of z denotes the number of each CuO_2 plane counting from the interface. Positive z is on LCO side while negative z is on LSCO side. $p(z)$ deviates from $x(z)$ and the difference between them is the loss/gain of charge density due to charge transfer. The comparison between (a) and (b) shows there is no fundamental difference between the two approaches.

Reviewer #2 (Remarks to the Author):

Comment 4: *The manuscript "Reentrance of interface superconductivity in a high- T_c cuprate heterostructure", by Jingyi Shen, Chuanyu Shi, Zhiming Pan, Lele Ju, Mingdong Dong, Gangfan Chen, Yichi Zhang, Congjun Wu, Yanwu Xie, and Jie Wu, contains result that are timely and interesting. However, before a definite statement about their relevance and soundness can be made, I ask the authors to consider the issues that are listed hereafter.*

Response 4: We are grateful for the Referee's positive assessment of our work and inspirational suggestions.

Comment 5: *Usually, the quantum confinement in the direction perpendicular to the interface produces a series of levels (that are the thresholds of two-dimensional sub-bands associated with the motion parallel to the interface). A discussion about the physics of the interface (in particular, about its superconducting properties) should attempt a connection with this sub-band structure and the hierarchy of the sub-band filling, when the density of carriers at the interface is varied.*

Response 5: Please see Response 3. In brief, the difference between the LCO/LSCO bilayers and LAO/STO (or KTO-related interfaces) is that the overdoped LSCO is metallic (though the effect of localization gets stronger as x approaches 1.0) while LAO and STO are both insulating. The missing of a bandgap at the LSCO side means the electrons are not confined at the LCO/LSCO interface.

Comment 6: *The solution of the Schrodinger-Poisson equations for oxide interfaces like LAO/STO or LTO/STO [PRL 116, 026804 (2016)] shows that the sub-band structure may be not rigid, and electron phase separation may occur, resulting in an inhomogeneous state at the interface. As I argue hereafter, I suspect that at least for some carrier density, the authors are observing the phenomenology of an inhomogeneous electron gas at the interface.*

Response 6: As already presented in Response 3 and 5, there is no quantum

confinement of charges at LCO/LSCO interface since overdoped LSCO is metallic. Nevertheless, we thank the Referee for pointing out the influence of inhomogeneity and we agree that a certain degree of inhomogeneity might be present in the LCO/LSCO bilayers.

Comment 7: *My main line of reasoning, when it comes to the homogeneity of the electron gas rests on the curves in Fig. 1a. The curves at 0.7 and 0.9 are so broad that their width cannot be possibly accounted for any reasonable fluctuational mechanism and rather points toward inhomogeneity [PRB 84, 014514 (2011)]. The fact that the resistance significantly drops, without seemingly reaching the zero-resistance state, likely indicates that a sizable superconducting cluster exists in the system, but does not percolate. The percolative character of the superconducting transition again seems to point toward inhomogeneity.*

Response 7: The $R(T)$ curves for $x = 0.7$ and 0.9 shows a lower T_c and a boarder transition. Though the superconducting transition of cuprates is generically broader than conventional BCS superconductors due to vortex excitations, we do agree that inhomogeneity very likely is at play behind the scenes. From the $R(T)$ curves of the LSCO, we can tell the localization due to scattering of disorders gets stronger as the doping x increases towards 1.0. These disorders might be inhomogeneous and caused by La/Sr randomness and oxygen vacancies, plus the spontaneous electron phase separation suggested by the Referee. We've added discussions on inhomogeneity and related references to the revised manuscript.

Comment 8: *The authors themselves write that their interfaces are structurally homogeneous, but admittedly cannot control oxygen vacancies. The argument that they should not have a direct effect on superconductivity is not compelling, my counterargument is that mobile oxygen vacancies may balance the inhomogeneous distribution of charge carriers at the interface, making electron phase separation easier. I point out that such an inhomogeneous electron gas need not be accompanied by a correspondingly inhomogeneous lattice configuration. The modulation of the electron charge may be mild (a tiny fraction of electron per unit cell), so that the lattice may be nearly unaffected.*

Response 8: Indeed, the mechanism proposed by the Referee is very inspiring and we'll pay attention to signatures of electron phase separation in our follow-up works. Discussions on this possibility have been added in the revised manuscript.

Comment 9: *I am also not convinced of the authors' discussion about the $\log(1/T)$ term in the resistivity. This phenomenology was found in underdoped cuprates long ago [PRL 77, 5417 (1996)], but it was convincingly argued that weak localization cannot possibly explain it [C. Castellani, P. Schwab, M. Grilli, (2002). On Localization Effects in Underdoped Cuprates. In: A. Bianconi, N. L. Saini (eds), Stripes and Related Phenomena. Selected Topics in Superconductivity, pp 361–367, vol 8. Springer, Boston, MA. https://doi.org/10.1007/0-306-47100-0_45]. The main argument is that weak localization gives a log contribution to the conductivity, to have a log contribution to the resistivity, it must be small so that a perturbative expansion can transform a log in the conductivity into a log in the resistivity. In underdoped cuprates, the log contribution is by no means small. Now, it would be somewhat surprising that there are two different mechanisms for the log contribution to the resistivity in bulk samples and in heterostructures. A thorough discussion of the points listed above is mandatory.*

Response 9: There is some similarity between $\rho(T)$ behaviors at lightly underdoped and heavily overdoped LSCO, including the $\log(1/T)$ contribution to $\rho(T)$. Actually, this motivated us first to investigate the possibility of whether a second superconducting dome exist at LSCO ($0.5 < x < 1.0$). Though the results for single layer LSCO so far are negative, instead we surprisingly discovered the reentrance of interface superconductivity that is the main findings of the current manuscript.

To exam whether the mechanism responsible for the $\log(1/T)$ term is identical for

the under- and over-doped side, we further measured the magneto-conductivity (MC) and found they are fundamentally different (Fig. R2): MC of overdoped LSCO ($x = 0.7$) shows characteristic “W” shape as a function of the B field at intermediate temperatures, e.g. 10 K, that is absent in MC of the underdoped LSCO ($x = 0.048$). Instead, the latter follows a dependence $\Delta\sigma \propto -B^2$ at $T > 15$ K, the temperature range at which $\log T$ contribution dominates $\sigma(T)$ for LSCO ($x = 0.048$) (Fig. R2a & R2b, reproduced from Physica C 404, 87–94 (2004)). In stark contrast, all the MC curves of LSCO ($x = 0.7$) can be well fitted by the expressions for weak localization (Fig. R2c & R2d). Thus, we conclude the $\log(1/T)$ term is due to the weak localization for overdoped LSCO. We are, however, open to other theoretical proposals as long as they are consistent with our experimental results. We thank the referee for the helpful information and have added the references and related discussion to the revised manuscript.

Fig. R2: The comparison between the resistivity and magneto-conductivity of under- and over-doped cuprates for several temperatures. a, The conductivity $\sigma(T)$ show $\log T$ contribution for underdoped LSCO single crystal. **b,** The corresponding magneto-conductivity $\Delta\sigma_{//} = \Delta\rho_{//}/\rho_0^2$ for LSCO ($x = 0.048$) at different T . Here $\Delta\rho_{//} = \rho - \rho_0$ and ρ_0 is the resistivity at zero field. [Panels a and b are reproduced from Physica C 404, 87–94 (2004)]. **c,** $\rho(T)$ of our overdoped LSCO ($x = 0.7$) thin film. **d,** The magneto-conductivity of LSCO ($x = 0.7$) $G = \frac{2\pi^2\hbar}{e^2}(\sigma(B) - \sigma(B = 0))$ shows a characteristic “W” shape at 10 K. $G(B)$ at every T that can be well fitted (red solid lines) by an expression $G = A \times \frac{3}{2} \left[\psi \left(\frac{1}{2} + \frac{B_1}{B} \right) - \ln \left(\frac{B_1}{B} \right) \right]$ derived from the weak localization theory [Ref. 46 of the manuscript].

Comment 10: There are also minor points:

a. The authors write “caution interdiffusion”, I suspect they mean “cation interdiffusion”.

b. The authors write “insert of Fig. 3d”, they probably mean “inset of Fig. 3d”.

Once the authors have responded to my criticism and complied with the issues I raised, I suggest that their

manuscript may be reconsidered for publication.

Response 10: We thank the Referee for pointing out our typos and they have been corrected in the revised manuscript. We've addressed the Referee's comments carefully and added discussions/clarifications as much as we could. We hope the improved manuscript will meet the Referee's standards.

Reviewer #3 (Remarks to the Author):

Comment 11: *The paper "Reentrance of interface superconductivity in a high-Tc cuprate heterostructure" by J. Y. Shen addresses an extremely important and interesting issue, namely superconductivity in cuprates in the high doping regime ($x > 0.8$). Specifically, the authors investigate interfacial superconductivity in the LSCO/LCO system, where they have successfully measured a fascinating phenomenon, namely the reentrance of superconductivity in the highly overdoped regime $0.8 < x < 1.0$. To explain the observed phenomenon, the authors propose a theoretical model based on charge transfer at the interface, which is a result of the interplay between the electrostatics of the junction determined by the work function in LSCO which changes with doping, and the charge localization effect in LSCO, experimentally confirmed by magnetconductance measurements. Undoubtedly, the research results contribute to a significant and novel direction in the study of high-temperature superconductivity.*

Response 11: We are very grateful for the Referee's encouraging comments and favorable recognition of the value of our work.

Comment 12: *Here are my comments and questions regarding the paper: Undoubtedly, the paper presents a very interesting experimental result, namely the reentrance of superconductivity in the range of high doping. Although the authors state in the introduction, "the reentrance of superconductivity has not been observed yet by tuning the carrier density in cuprate superconductor.", I would like to point out that a similar phenomenon was observed in 2016 for a monoatomic CuO₂ layer deposited on a Bi₂Sr₂CaCu₂O_{8+δ} substrate (CuO₂/BSCCO) [Sci. Bull. 61, 1239 (2016)]. However, it does seem that these are indeed the two papers in which superconductivity has been observed in cuprates in the highly overdoped regime. Therefore, I consider the experimental result presented in this paper to be highly significant.*

Response 12: The superconductivity at higher dopings outside of the well-known superconducting dome is a very attractive possibility worth of explorations. This in fact motivated us to study the extremely overdoped LSCO ($0.5 < x < 1.0$). While the single layer LSCO is non-superconducting, somewhat disappointing, we surprisingly found the LCO/LSCO bilayer manifests the reentrance of interface superconductivity. As for the CuO₂/BSCCO, it is argued that the superconductivity of CuO₂ is induced by interfacial charge transfer, similar to the LCO/LSCO bilayer. While we believe the dopings of CuO₂ must be very high as DFT calculations showed [Phys. Rev. B 103, 144511 (2021); Phys. Rev. Lett. 121, 227002 (2018)] (although no direct measurement of its doping level is available in the literature to the best of our knowledge), clear experimental evidence on the reentrance of superconductivity or the two-dome structure with chemical doping in this material has not been reported yet since it lacks the studies of CuO₂/BSCCO superconductivity on the doping level of BSCCO. In Response 14, we made a detailed comparison between CuO₂/BSCCO and our LCO/LSCO.

Comment 13: *However, in my opinion, the theoretical model explaining the observed phenomenon is too simplified. In fact, it reduces to the standard procedure of determining the built-in potential in junctions, namely the balance between diffusion current and drift current. However, I do not fully understand how this model accounts for the localization and the change in its strength with increasing doping, which the authors mention when analyzing the resistance measurements. It seems much more reasonable in this regard to consider a*

full Schrödinger-Poisson model limited to a few relevant bands that participate in charge redistribution at the junction. Furthermore, I would like to point out that the entire paper lacks a model describing superconductivity and the critical temperature which in the paper is described by the empirical formula.

Response 13: Please see Response 3 for detailed explanations. There we've provided the results from the Schrödinger-Poisson model and showed it qualitatively agrees with our previous model. In brief, the interface superconductivity resides at a single CuO_2 plane located at the LCO side close to the LCO/LSCO interface (see Response 16 for details). Thus, superconductivity is generated by doping the undoped LCO with transferred charges from LSCO. It conforms to the same mechanism giving rise to high temperature superconductivity (HTS). Since the mechanism of HTS is still a major unresolved problem in physics, we don't attempt to address it in this work and merely take it as an established fact that the superconducting T_c and the carrier density p in LCO has an empirical relation $T_c = 3200 \times (p - 0.06) \times (0.26 - p)$. This in fact is a pretty common practice in the HTS field as some researchers actually use this relation to calculate the actual doping levels in samples from the measured T_c values [For instance, see Phys. Rev. B 83, 054506 (2011)].

As for the effect of charge localization, we observed an increase of localization strength as the doping x increases for the single layer LSCO, which results in a transition from weak localization to strong localization (VRH hopping) at $x \sim 0.8$. Based on this experimental fact, we can tell the density of pinning centers responsible for localization must increase with x . This leads to the consequence that the density of mobile carriers decreases with x and completely diminishes at $x > 0.8$. Thus, the charge transfer is severely suppressed around $x = 0.8$. For $x > 0.8$, all the carriers in LSCO are localized and the hopping of holes from LSCO to LCO is possible only when the difference in work functions ΔW between LCO and LSCO is bigger than the potential well depth V of the localization centers. Since ΔW increases with x (Fig. 4a), the interfacial charge transfer revives when $\Delta W > \bar{V}$ here \bar{V} is the average potential well depth. This gives a natural explanation of the reentrance of interface superconductivity. Detailed calculations can be found in the Supplementary Information.

Comment 14: In the paper, the reentrance effect of superconductivity was explained based on charge transfer blockade for $x=0.8$. However, it seems that the presented explanation may not be the only valid one in this case. The considered system is somewhat similar to $\text{CuO}_2/\text{BSCCO}$ [Sci. Bull. 61, 1239 (2016)], where increased doping leads to the appearance of a new band (Lifshitz transition), which is associated with the emergence of superconductivity. A comprehensive theoretical model of the superconducting state for this system was presented in Phys. Rev. B 103, 144511 (2021), where it was clearly shown that around $x=0.8$, a Lifshitz transition occurs (an additional $d_{x^2-y^2}$ band crosses the Fermi level), and the induced superconductivity has the extended s -wave symmetry. It should be noted that in the considered paper, a certain change in the electronic structure is observed for LSCO with increasing x - the sign change of the Hall resistance is observed, which the authors suggest may be evidence of a Fermi surface modification. Taking into account a global trend that occurs in the phase diagram of doped cuprates, it can be expected that the phenomenon of superconductivity emergence in the range of high doping described in Phys. Rev. B 103, 144511 (2021) also takes place in the LCO/LSCO system, especially considering that the doping ranges where superconductivity is observed in both papers are the same. Therefore, from a theoretical standpoint, it would be necessary to perform DFT calculations for LCO/LSCO to investigate the interface's influence on the energetic distribution of bands and create a model of superconductivity based on such calculations. Here, I would like to mention that the interface can significantly alter the electronic structure of a material, as evidenced by the 50 meV lowering of the d_{xy} bands in the LAO/STO system. In conclusion, it appears that a thorough analysis of the electronic structure in LCO/LSCO would be necessary to create a complete model of the described phenomenon and confirm the feasibility of the proposed model in the paper.

Response 14: We thank the referee for thoughtful suggestions and inspirational proposals. It would be an even more exciting discovery if the interface superconductivity in LSCO($0.8 < x \leq 1.0$)/LCO is similar to that in $\text{CuO}_2/\text{BSCCO}$ so that it might be s-wave symmetry and related with $d(3z^2-r^2)$ band. In this scenario, the reentrance of interface superconductivity would be caused by the transition from d-wave symmetry and $d(x^2-y^2)$ band to s-wave symmetry and $d(3z^2-r^2)$ band. While this hypothesis seems very attractive to us, we feel it is hard to be reconciled with the following experimental facts:

1. For the single layer LSCO($0 < x \leq 1.0$), we observe only one superconducting dome within $0.06 \leq x \leq 0.26$ and we haven't found a second dome at higher doping levels.
2. As detailed in Response 3 and Response 15, the interface superconductivity of LCO/LSCO resides at the LCO side and the main role of the LSCO layer is to provide transferred charges. Thus, superconductivity is due to the doping of LCO layer, the same as the single layer LSCO.
3. As the Schrödinger-Poisson model and our original Poisson model shows, together with previous studies in the literature [Ref. 15, 25-27 of the manuscript], the transferred carrier density at each CuO_2 plane of the LCO side decays with the distance from the interface (Fig. R1). Every CuO_2 plane with the right amount of carrier density superconducts and T_c of the bilayer is determined by the highest T_c among these CuO_2 planes.
4. Whether or not there exists a CuO_2 plane at the LCO side, e.g. $z = 1$, with high carrier density ($p \gg 0.26$) and superconducts like CuO_2 in $\text{CuO}_2/\text{BSCCO}$, other CuO_2 planes, e.g. $z = 2, 3$, etc. will superconduct if the carrier density there $p(z)$ falls within $0.06 \leq p \leq 0.26$. So the demise of superconductivity at LSCO($x = 0.8$)/LCO cannot be related to the hypothesis.
5. Our results show T_c of LCO/LSCO($0.45 \leq x \leq 1.0$) never exceeds the maximum T_c of the single layer LSCO within the superconducting dome, challenging the idea that there are two different mechanisms responsible for the observed T_c .

From the above points, we feel that the mechanism underlying interface superconductivity in our LCO/LSCO is different from that in $\text{CuO}_2/\text{BSCCO}$ and it presumably requires other conditions, e.g. epitaxial tension, surface reconstruction, or removal of the top La-O plane, to reproduce the phenomenon in $\text{CuO}_2/\text{BSCCO}$. Nevertheless, it is an intriguing comparison between these two systems, and we've added this discussion and related references to the revised manuscript.

***Comment 15:** It seems that a measurement of local density of states could serve as confirmation or refutation of the theoretical predictions from the PRB paper. In the case of extended s-wave superconductivity, we should observe a characteristic U-shaped profile, in contrast to the d-wave symmetry characterized by the V-shaped local density of states. Are such additional measurements possible in the proposed system?*

Response 15: It's a splendid suggestion but the superconducting CuO_2 plane is unfortunately located at the LCO side right next to the LCO/LSCO interface. There is only one CuO_2 plane superconducting with highest T_c (see Response 16) and it is buried beneath all other LCO layers while some of these LCO layers are doped with holes transferred across the interface. This makes a direct measurement of the local density of states of the superconducting CuO_2 plane extremely difficult, if not impossible.

Comment 16: A very basic question: Where is the superconductivity located? Is there a thin copper-oxygen layer near the interface on the LSCO side? Or is it rather located on the LCO side where the charge is transferred?

Response 16: This indeed is a very fundamental question central to research on interface superconductivity. Luckily, experimentalists, we included, have innovated a neat experimental method to address this question.

As a known fact, the replacement of Cu by Zn, even as little as 3%, can suppress HTS very effectively. The state-of-the-art molecular beam epitaxy (MBE) or pulsed laser deposition (PLD) technique enables us to choose a single CuO_2 plane and selectively do the Zn replacement (δ -doping) during film synthesis. A series of samples were synthesized with Zn doping location varying from $N = 1$ to 6 at LCO side to $N = -1$ to -6 at LSCO side. The results clearly showed that Zn doping at $N = 2$ CuO_2 plane significantly suppresses T_c as compared to other N (Fig. R3a and R3b). This illustrates for the LCO/LSCO($x = 0.45$) bilayer, the interface superconductivity resides at $N = 2$ CuO_2 plane. The above data and conclusion are from Ref. 14 of the manuscript.

For our LCO/LSCO films, we tried something similar. To save the efforts and reduce the complexity during film synthesis, we replaced 5% of Cu by Fe for either the entire LSCO or LCO layer (Fig. R3c). The results unambiguously show the interface superconductivity of LCO/LSCO($x = 1.0$) is at the LCO side.

Fig. R3: The δ -doping experiment. **a**, A schematic drawing showing we can selectively replace 3% of Cu at a particular CuO_2 plane (highlighted in yellow) by Zn in the LCO/LSCO bilayer. The number N represents the counting of CuO_2 planes from the interface with positive value for LCO side and negative for LSCO side. **b**, The normalized $R(T)$ curves unambiguously show only doping $N = 2$ CuO_2 plane with Zn can effectively suppress T_c , in stark contrast to all other $N \neq 2$ results. This illustrates the interface superconductivity resides at the $N = 2$ CuO_2 plane at LCO side. [Panels a, b are reproduced from Ref. 14 of the manuscript]. **c**, We selectively doped LCO or LSCO layer with 5% Fe. Compared with the original LCO/LSCO($x = 1.0$) bilayer (black curve), LCO/LSCO(Fe) remains roughly the

same (blue curve) while LCO/LSCO(Fe) (red curve) loses its superconductivity completely. This clearly shows the interface superconductivity of LCO/LSCO is at the LCO side.

***Comment 17:** In conclusion, while I find the experimental results interesting and significant for the field of high-temperature superconductors, I would recommend undertaking substantial efforts to develop a realistic theoretical model that adequately explains the observed phenomenon.*

Response 17: We've carefully addressed the comments raised by the referees and revised the manuscript accordingly. We are certain that these improvements enhance the quality of our work, and we thank the Referee very much for devoted efforts.

REVIEWERS' COMMENTS

Reviewer #1 (Remarks to the Author):

The comments by this reviewer were addressed satisfactorily. I recommend publication.

Reviewer #2 (Remarks to the Author):

The manuscript "Reentrance of interface superconductivity in a high- T_c cuprate heterostructure", by J. Y. Shen, et al., has been revised along the lines suggested by the reviewers. Overall, I find that the authors' reply to the various comments and issues raised by the reviewers is fair. The authors now state in a much clearer manner their assumptions and the possible limitations to their interpretation, so that the reader can now judge not only about the reliability of the experimental results, but also about the soundness of the proposed scenario.

I think that there is still much to say about the role of inhomogeneity, and about the mechanism that leads to the log term in the resistance, but this may reasonably be beyond the reach of the present study.

Maybe, but this is not mandatory, a more cautious statement about the possible role of oxygen vacancies, together with other sources of inhomogeneity, might be made in the Result and discussion section, when discussing the effect of oxygen vacancies. As I said, my (simple-minded) interpretation of what is going on in the sheet-resistance data is that the superconducting transition is acquiring a percolative character, strictly speaking there is a whole range of x where the systems does not exhibit a zero-resistance state down to $T=0$, so, for reasons that may interplay with the scenario proposed by the authors, we are facing a lack of percolation. If the would be percolative transition temperature becomes very negative, whatever the reason, the corresponding sheet-resistance curve would not exhibit hints of a suppression at all $T \geq 0$.

As a minor remark, the sentence "... is mainly due to not the weak localization but the reduction ..." seems to be ill-formulated, a correct formulation, if I understand correctly what the authors mean here, is "... is mainly due not to the weak localization but to the reduction ...".

With the above changes made, I suggest that the present piece of work may now be accepted for publication.

Reviewer #3 (Remarks to the Author):

I would like to thank the authors for a very extensive and comprehensive answers to my questions. Most of them satisfied my curiosity as a reviewer and answered any doubts I had. Particularly important seems to be the additional calculations performed by the authors in the Schrodinger-Poisson model and the experiment showing the localization of superconductivity in the considered system. Despite this, I think that the arguments on the basis of which the authors reject the proposed hypothesis, according to which superconductivity for $x > 0.8$ arises as a result of the appearance of the new band, are not entirely accurate. The fact that a single LSCO layer does not exhibit a second superconducting dome cannot be an argument that the LCO/LSCO heterostructure does not change its electronic structure. Similarly, the argument about the value of T_c does not prove that the nature of both superconducting domes is the same. Nevertheless, I understand that performing a full DFT calculation is well beyond the scope of this paper, and I wish to emphasize that all measurements presented in the paper indicate that the mechanism proposed by the authors is probably the correct one. Nevertheless, my inquisitiveness tells me that until we clearly show in calculations or experiments that the superconductivity at $x > 0.8$ does not arise as a result of the presence of a new band, this scenario is possible, especially since the considered system is significantly similar to the one considered in the paper Phys. Rev. B 103, 144511. (2021). Anyway, I think it's good for the paper that there is now a mention of the results for the CuO₂ monolayer.

Taking into account the authors answers and the fact that the article undoubtedly constitutes an important voice in a debate of superconductivity at high T_c interfaces, I would like to recommend the papers in the present form for publication in Nature Communications.

Point-by-point response to the referees' comments.

Reviewer #1 (Remarks to the Author):

Comment 1: The comments by this reviewer were addressed satisfactorily. I recommend publication.

Response 1: We thank the Referee very much for the devoted efforts and valuable suggestions.

Reviewer #2 (Remarks to the Author):

Comment 2: The manuscript "Reentrance of interface superconductivity in a high-T_c cuprate heterostructure", by J. Y. Shen, et al., has been revised along the lines suggested by the reviewers. Overall, I find that the authors' reply to the various comments and issues raised by the reviewers is fair. The authors now state in a much clearer manner their assumptions and the possible limitations to their interpretation, so that the reader can now judge not only about the reliability of the experimental results, but also about the soundness of the proposed scenario.

Response 2: We are grateful that the Referee appreciates our endeavors in improving the manuscript. Indeed, we feel the paper is in a much better shape thanks to all the Referees' constructive suggestions.

Comment 3: I think that there is still much to say about the role of inhomogeneity, and about the mechanism that leads to the log term in the resistance, but this may reasonably be beyond the reach of the present study.

Response 3: We agree sincerely that these issues deserve further investigations. However, as the Referee already pointed out, these contents are out of the scope of the current paper. We are collecting more data on the single layer LSCO ($0.45 \leq x \leq 1.0$) and hopefully will be able to address these issues in our follow-up works.

Comment 4: Maybe, but this is not mandatory, a more cautious statement about the possible role of oxygen vacancies, together with other sources of inhomogeneity, might be made in the Result and discussion section, when discussing the effect of oxygen vacancies. As I said, my (simple-minded) interpretation of what is going on in the sheet-resistance data is that the superconducting transition is acquiring a percolative character, strictly speaking there is a whole range of x where the systems does not exhibit a zero-resistance state down to $T=0$, so, for reasons that may interplay with the scenario proposed by the authors, we are facing a lack of percolation. If the would be percolative transition temperature becomes very negative, whatever the reason, the corresponding sheet-resistance curve would not exhibit hints of a suppression at all $T \geq 0$.

Response 4: This comment is inspirational and a brief discussion on the role of percolation is now added in the Result and discussion section of the revised manuscript.

Comment 5: As a minor remark, the sentence "... is mainly due to not the weak localization but the reduction ..." seems to be ill-formulated, a correct formulation, if I understand correctly what the authors mean here, is "... is mainly due not to the weak localization but to the reduction ...".

Response 5: We thank the Referee for helping us correct the typos. This now has been corrected in the revised manuscript.

Comment 6: With the above changes made, I suggest that the present piece of work may now be accepted for publication.

Response 6: The Referee's comments are very professional and constructive. For

that, we are very grateful.

Reviewer #3 (Remarks to the Author):

Comment 7: *I would like to thank the authors for a very extensive and comprehensive answers to my questions. Most of them satisfied my curiosity as a reviewer and answered any doubts I had. Particularly important seems to be the additional calculations performed by the authors in the Schrodinger-Poisson model and the experiment showing the localization of superconductivity in the considered system.*

Response 7: It's encouraging to know that the Referee appreciates the improvements we've implemented based on all the Referees' comments.

Comment 8: *Despite this, I think that the arguments on the basis of which the authors reject the proposed hypothesis, according to which superconductivity for $x > 0.8$ arises as a result of the appearance of the new band, are not entirely accurate. The fact that a single LSCO layer does not exhibit a second superconducting dome cannot be an argument that the LCO/LSCO heterostructure does not change its electronic structure. Similarly, the argument about the value of T_c does not prove that the nature of both superconducting domes is the same. Nevertheless, I understand that performing a full DFT calculation is well beyond the scope of this paper, and I wish to emphasize that all measurements presented in the paper indicate that the mechanism proposed by the authors is probably the correct one. Nevertheless, my inquisitiveness tells me that until we clearly show in calculations or experiments that the superconductivity at $x > 0.8$ does not arise as a result of the presence of a new band, this scenario is possible, especially since the considered system is significantly similar to the one considered in the paper Phys. Rev. B 103, 144511. (2021). Anyway, I think it's good for the paper that there is now a mention of the results for the CuO₂ monolayer.*

Response 8: We think the Referee's above judgement is fair. In accordance with this, we revised the manuscript by mentioning briefly the hypothesis of emergence of a new band and leaving this possibility open for further investigations.

Comment 9: *Taking into account the authors answers and the fact that the article undoubtedly constitutes an important voice in a debate of superconductivity at high T_c interfaces, I would like to recommend the papers in the present form for publication in Nature Communications.*

Response 9: We are very pleased that our paper reaches the high standards of the Referee and the journal.